# A Meta-Analysis of Overfitting in Machine Learning

**Rebecca Roelofs**[*]
UC Berkeley
roelofs@berkeley.edu

**Sara Fridovich-Keil**[*]
UC Berkeley
sfk@berkeley.edu

**John Miller**
UC Berkeley
miller_john@berkeley.edu

**Vaishaal Shankar**
UC Berkeley
vaishaal@berkeley.edu

**Moritz Hardt**
UC Berkeley
hardt@berkeley.edu

**Benjamin Recht**
UC Berkeley
brecht@berkeley.edu

**Ludwig Schmidt**
UC Berkeley
ludwig@berkeley.edu

## Abstract

We conduct the first large meta-analysis of overfitting due to test set reuse in the machine learning community. Our analysis is based on over one hundred machine learning competitions hosted on the Kaggle platform over the course of several years. In each competition, numerous practitioners repeatedly evaluated their progress against a holdout set that forms the basis of a public ranking available throughout the competition. Performance on a separate test set used only once determined the final ranking. By systematically comparing the public ranking with the final ranking, we assess how much participants adapted to the holdout set over the course of a competition. Our study shows, somewhat surprisingly, little evidence of substantial overfitting. These findings speak to the robustness of the holdout method across different data domains, loss functions, model classes, and human analysts.

## 1   Introduction

The holdout method is central to empirical progress in the machine learning community. Competitions, benchmarks, and large-scale hyperparameter search all rely on splitting a data set into multiple pieces to separate model training from evaluation. However, when practitioners repeatedly reuse holdout data, the danger of overfitting to the holdout data arises [6, 13].

Despite its importance, there is little empirical research into the manifested robustness and validity of the holdout method in practical scenarios. Real-world use cases of the holdout method often fall outside the guarantees of existing theoretical bounds, making questions of validity a matter of guesswork.

Recent replication studies [16] demonstrated that the popular CIFAR-10 [10] and ImageNet [5, 18] benchmarks continue to support progress despite years of intensive use. The longevity of these benchmarks perhaps suggests that overfitting to holdout data is less of a concern than reasoning from first principles might have suggested. However, this is evidence from only two, albeit important, computer vision benchmarks. It remains unclear whether the observed phenomenon is specific to the data domain, model class, or practices of vision researchers. Unfortunately, these replication studies

---

[*]Equal contribution

required assembling new test sets from scratch, resulting in a highly labor-intensive analysis that is difficult to scale.

In this paper, we empirically study holdout reuse at a significantly larger scale by analyzing data from 120 machine learning competitions on the popular Kaggle platform [2]. Kaggle competitions are a particularly well-suited environment for studying overfitting since data sources are diverse, contestants use a wide range of model families, and training techniques vary greatly. Moreover, Kaggle competitions use public and private test data splits which provide a natural experimental setup for measuring overfitting on various datasets.

To provide a detailed analysis of each competition, we introduce a coherent methodology to characterize the extent of overfitting at three increasingly fine scales. Our approach allows us both to discuss the overall "health" of a competition across all submissions and to inspect signs of overfitting separately among the top submissions. In addition, we develop a statistical test specific to the classification competitions on Kaggle to compare the submission scores to those arising in an ideal null model that assumes no overfitting. Observed data that are close to data predicted by the null model is strong evidence against overfitting.

Overall, we conclude that the classification competitions on Kaggle show little to no signs of overfitting. While there are some outlier competitions in the data, these competitions usually have pathologies such as non-i.i.d. data splits or (effectively) small test sets. Among the remaining competitions, the public and private test scores show a remarkably good correspondence. The picture becomes more nuanced among the highest scoring submissions, but the overall effect sizes of (potential) overfitting are typically small (e.g., less than 1% classification accuracy). Thus, our findings show that substantial overfitting is unlikely to occur naturally in regular machine learning workflows.

## 2 Background and setup

Before we delve into the analysis of the Kaggle data, we briefly define the type of overfitting we study and then describe how the Kaggle competition format naturally lends itself to investigating overfitting in machine learning competitions.

### 2.1 Adaptive overfitting

"Overfitting" is often used as an umbrella term to describe any unwanted performance drop of a machine learning model. Here, we focus on *adaptive* overfitting, which is overfitting caused by test set reuse. While other phenomena under the overfitting umbrella are also important aspects of reliable machine learning (e.g. performance drops due to distribution shifts), they are beyond the scope of our paper since they require an experimental setup different from ours.

Formally, let $f : \mathcal{X} \to \mathcal{Y}$ be a trained model that maps examples $x \in \mathcal{X}$ to output values $y \in \mathcal{Y}$ (e.g., class labels or regression targets). The standard approach to measuring the performance of such a trained model is to define a loss function $L : \mathcal{Y} \times \mathcal{Y} \to \mathbb{R}$ and to draw samples $S = \{(x_1, y_1), \ldots, (x_n, y_n)\}$ from a data distribution $\mathcal{D}$ which we then use to evaluate the test loss $L_S(f) = \sum_{i=1}^{n} L(f(x_i), y_i)$. As long as the model $f$ does not depend on the test set $S$, standard concentration results [19] show that $L_S(f)$ is a good approximation of the true performance given by the population loss $L_{\mathcal{D}}(f) = \mathbb{E}_{\mathcal{D}}[L(f(x), y)]$.

However, machine learning practitioners often undermine the assumption that $f$ does not depend on the test set by selecting models and tuning hyperparameters based on the test loss. Especially when algorithm designers evaluate a large number of different models on the same test set, the final classifier may only perform well on the specific examples in the test set. The failure to generalize to the entire data distribution $\mathcal{D}$ manifests itself in a large *adaptivity gap* $L_{\mathcal{D}}(f) - L_S(f)$ and leads to overly optimistic performance estimates.

### 2.2 Kaggle

Kaggle is the most widely used platform for machine learning competitions, currently hosting 1,461 active and completed competitions. Various organizations (companies, educators, etc.) provide the

datasets and evaluation rules for the competitions, which are generally open to any participant. Each competition is centered around a dataset consisting of a training set and a test set.

Considering the danger of overfitting to the test set in a competitive environment, Kaggle subdivides each test set into public and private components. The subsets are randomly shuffled together and the entire test set is released without labels, so that participants should not know which test samples belong to which split. Hence participants submit predictions for the *entire* test set. The Kaggle server then internally evaluates each submission on both public and private splits and updates the public competition leaderboard only with the score on the public split. At the end of the competition, Kaggle releases the private scores, which determine the winner.

Kaggle has released the MetaKaggle dataset[2], which contains detailed information about competitions, submissions, etc. on the Kaggle platform. The structure of Kaggle competitions makes MetaKaggle a useful dataset for investigating overfitting empirically at a large scale. In particular, we can view the public test split $S_{\text{public}}$ as the regular test set and use the held-out private test split $S_{\text{private}}$ to approximate the population loss. Since Kaggle competitors do not receive feedback from $S_{\text{private}}$ until the competition has ended, we assume that the submitted models may be overfit to $S_{\text{public}}$ but not to $S_{\text{private}}$.[3] Under this assumption, the difference between private and public loss $L_{S_{\text{private}}}(f) - L_{S_{\text{public}}}(f)$ is an approximation of the adaptivity gap $L_{\mathcal{D}}(f) - L_S(f)$. Hence our setup allows us to estimate the amount of overfitting occurring in a typical machine learning competition. In the rest of this paper, we will analyze the public versus private score differences as a proxy for adaptive overfitting.

Due to the large number of competitions on the Kaggle platform, we restrict our attention to the most popular classification competitions. In particular, we survey the competitions with at least 1,000 submissions before the competition deadline. Moreover, we include only competitions with evaluation metrics that have at least 10 competitions. These metrics are classification accuracy, AUC (area under curve), MAP@K (mean average precision), the logistic loss, and a multiclass variant of the logistic loss. Appendix A provides more details about our selection criteria.

## 3  Detailed analysis of competitions scored with classification accuracy

We begin with a detailed look at classification competitions scored with the standard accuracy metric. Classification is the prototypical machine learning task and accuracy is a widely understood performance measure. This makes the corresponding competitions a natural starting point to understand nuances in the Kaggle data. Moreover, there is a large number of accuracy competitions, which enables us to meaningfully compare effect sizes across competitions. As we will see in Section 3.3, accuracy competitions also offer the advantage that we can obtain measures of statistical uncertainty. Later sections will then present an overview of the competitions scored with other metrics.

We conduct our analysis of overfitting at three levels of granularity that become increasingly stringent. The first level considers all submissions in a competition and checks for systematic overfitting that would affect a substantial number of submissions (e.g., if public and private score diverge early in the competition). The second level then zooms into the top 10% of submissions (measured by public accuracy) and conducts a similar comparison of public to private scores. The goal here is to understand whether there is more overfitting among the best submissions since they are likely most adapted to the test set. The third analysis level then takes a mainly quantitative approach and computes the probabilities of the observed public vs. private accuracy differences under an ideal null model. This allows us to check if the observed gaps are larger than purely random fluctuations.

In the following subsections, we will apply these three analysis methods to investigate four accuracy competitions. These four competitions are the accuracy competitions with the largest number of submissions and serve as representative examples for a typical accuracy competition in the MetaKaggle dataset (see Table 1 for information about these competitions). Section 3.4 then complements these analyses with a quantitative look at all competitions before we summarize our findings for accuracy competitions in Section 3.5.

Table 1: The four accuracy competitions with the largest number of submissions. $n_{\text{public}}$ is the size of the public test set and $n_{\text{private}}$ is the size of the private test set.

| ID | Name | # Submissions | $n_{\text{public}}$ | $n_{\text{private}}$ |
|---|---|---|---|---|
| 5275 | Can we predict voting outcomes? | 35,247 | 249,344 | 249,343 |
| 3788 | Allstate Purchase Prediction Challenge | 24,532 | 59,657 | 139,199 |
| 7634 | TensorFlow Speech Recognition Challenge | 24,263 | 3,171 | 155,365 |
| 7115 | Cdiscount's Image Classification Challenge | 5,859 | 53,0455 | 1,237,727 |

### 3.1 First analysis level: visualizing the overall trend

As mentioned in Section 2.2, the main quantities of interest are the accuracies on the public and private parts of the test set. In order to visualize this information at the level of all submissions to a competition, we create a scatter plot of the public and private accuracies. In an ideal competition with a large test set and without any overfitting, the public and private accuracies of a submission would all be almost identical and lie near the $y = x$ diagonal. On the other hand, substantial deviations from the diagonal would indicate gaps between the public and private accuracy and present possible evidence of overfitting in the competition.

Figure 1 shows such scatter plots for the four accuracy competitions mentioned above. In addition to a point for each submission, the scatter plots also contain a linear regression fit to the data. All four plots show a linear fit that is close to the $y = x$ diagonal. In addition, three of the four plots show very little variation around the diagonal. The variation around the diagonal in the remaining competition is largely symmetric, which indicates that it is likely the effect of random chance.

These scatter plots can be seen as indicators of overall competition "health": in case of pervasive overfitting, we would expect a plateauing trend where later points mainly move on the $x$-axis (public accuracy) but stagnate on the $y$-axis (private accuracy). In contrast, the four plots in Figure 1 show that as submissions progress on the public test set, they see corresponding improvements also on the private test set. Moreover, the public scores remain representative of the private scores.

As can be seen in Appendix B.3, this overall trend is representative for the 34 accuracy competitions. All except one competition show a linear fit close to the main diagonal. The only competition with a substantial deviation is the "TAU Robot Surface Detection" competition (ID 12598). We contacted the authors of this competition and confirmed that there are subtleties in the public / private split which undermine the assumption that the two splits are i.i.d. Hence we consider this competition to be an outlier since it does not conform to the experimental setup described in Section 2.2. So at least on the coarse scale of the first analysis level, there are little to no signs of adaptive overfitting: it is easier to make genuine progress on the data distribution in these competitions than to substantially overfit to the test set.

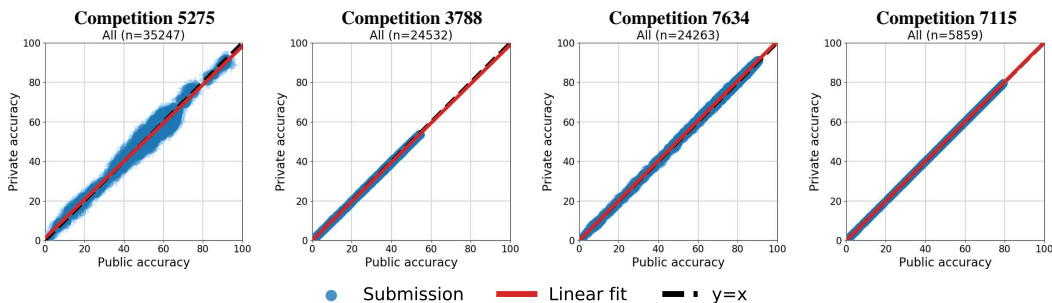

Figure 1: Private versus public accuracy for all submissions in the most popular Kaggle accuracy competitions. Each point corresponds to an individual submission (shown with 95% Clopper-Pearson confidence intervals, although the confidence intervals are smaller than the plotted data points).

## 3.2 Second analysis level: zooming in to the top submissions

While the scatter plots discussed above give a comprehensive picture of an entire competition, one concern is that overfitting may be more prevalent among the submissions with the highest public accuracy since they may be more adapted to the public test set. Moreover, the best submissions are those where overfitting would be most serious since invalid accuracies there would give a misleading impression of performance on future data. So to analyze the best submissions in more detail, we also created scatter plots for the top 10% of submissions (as scored by public accuracy).

Figure 2 shows scatter plots for the same four competitions as before. Since the axes now encompass a much smaller range (often only a few percent), they give a more nuanced picture of the performance among the best submissions.

In the leftmost and rightmost plot, the linear fit for the submissions still closely tracks the $y = x$ diagonal. Hence there is little sign of overfitting also among the top 10% of submissions. On the other hand, the middle two plots in Figure 2 show noticeable deviations from the main diagonal. Interestingly, the linear fit is *above* the diagonal in Competition 7634 and *below* the main diagonal in Competition 3788. The trend in Competition 3788 is more concerning since it indicates that the public accuracy overestimates the private accuracy. However, in both competitions the absolute effect size (deviation from the diagonal) is small (about 1%). It is also worth noting that the accuracies in Competition 3788 are not in a high-accuracy regime but around 55%. So the relative error from public to private test set is small as well.

Appendix B.4 contains scatter plots for the remaining accuracy competitions that show a similar overall trend. Besides the Competition 12598 discussed in the previous subsection, there are two additional competitions with a substantial public vs. private deviation. One competition is #3641, which has a total test set size of about 4,000 but is only derived from 7 human test subjects (the dataset consists of magnetoencephalography (MEG) recordings from these subjects). The other competition is 12681, which contains a public test set of size 90. Very small (effective) test sets make it easier to reconstruct the public / private split (and then to overfit), and also make the public and private scores more noisy. Hence we consider these two competitions to be outliers. Overall, the second analysis level shows that if overfitting occurs among the top submissions, it only does so to a small extent.

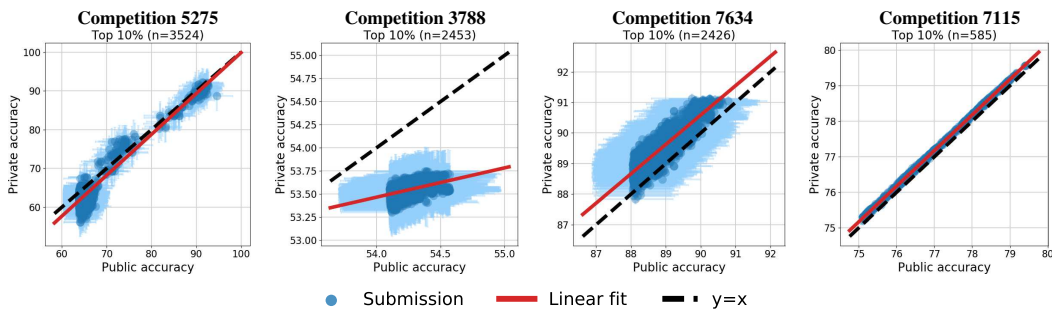

Figure 2: Private versus public accuracy for the top 10% of submissions in the most popular Kaggle accuracy competitions. Each point corresponds to an individual submission (shown with 95% Clopper-Pearson confidence intervals).

## 3.3 Third analysis level: quantifying the amount of random variation

When discussing deviations from the ideal $y = x$ diagonal in the previous two subsections, an important question is how much variation we should expect from random chance. Due to the finite sizes of the test sets, the public and private accuracies will never match exactly and it is a priori unclear how much of the deviation we can attribute to random chance and how much to overfitting.

To quantitatively understand the expected random variation, we compute the probability of a given public vs. private deviation (*p-value*) for a simple null model. By inspecting the distribution of the resulting p-values, we can then investigate to what extent the observed deviations can be attributed to random chance.

We consider the following null model under which we compute p-values for observing certain gaps between the public and private accuracies. We fix a submission that makes a given number of mistakes

on the *entire* test set (public and private split combined). We then randomly split the test set into two parts with sizes corresponding to the public and private splits of the competition. This leads to a certain number of mistakes (and hence accuracy) on each of the two parts. The p-value for this submission is then given by the probability of the event

$$|\text{public\_accuracy} - \text{private\_accuracy}| \geq \varepsilon \qquad (1)$$

where $\varepsilon$ is the observed deviation between public and private accuracy. We describe the details of computing these p-values in Appendix B.5.

Figure 3 plots the distribution of these p-values for the same four competitions as before. To see potential effects of overfitting, we show p-values for all submissions, the top 10% of submissions (as in the previous subsection), and for the first submission for each team in the competition. We plot the first submissions separately since they should be least adapted to the test set and hence show the smallest amount of overfitting.

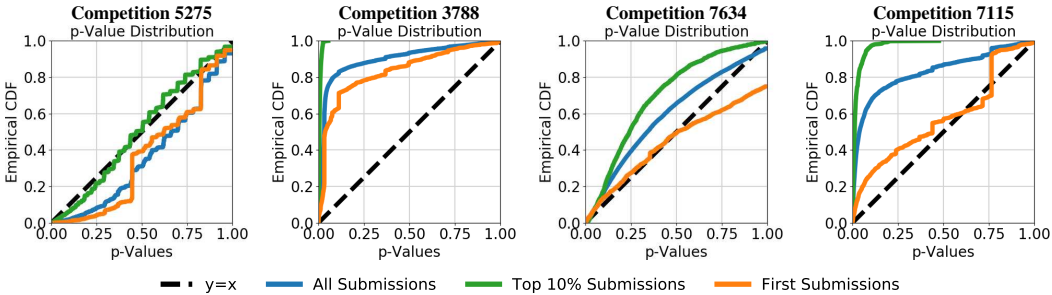

Figure 3: CDFs of the p-values for three (sub)sets of submissions in the four accuracy competitions with the largest number of submissions.

Under our ideal null hypothesis, the p-values would have a uniform distribution with a CDF following the $y = x$ diagonal. However, this is only the case for Competition 5275 and the first submissions in Competitions 7634 & 7115, and even there only approximately. Most other curves show a substantially larger number of small p-values (large gaps between public and private accuracy) than expected under the null model. Moreover, the top 10% of submissions always exhibit the smallest p-values, followed by all submissions and then the first submissions.

When interpreting these p-value plots, we emphasize that the null model is highly idealized. In particular, we assume that every submission is evaluated on its own *independent* random public / private split, which is clearly not the case for Kaggle competitions. So for correlated submissions, we would expect clusters of approximately equal p-values that are unlikely to arise under the null model. Since it is plausible that many models are trained with standard libraries such as XGBoost [4] or scikit-learn [14], we conjecture that correlated models are behind the jumps in the p-value CDFs.

In addition, it is important to note that for large test sets (e.g., the 198,856 examples in Competition 3788), even very small systematic deviations between the public and private accuracies are statistically significant and hence lead to small p-values. So while the analysis based on p-value plots does point towards irregularities such as overfitting, the overall effect size (deviation between public and private accuracies) can still be small.

Given the highly discriminative nature of the p-values, a natural question is whether any competition exhibits approximately uniform p-values. As can be seen in Appendix B.5, some competitions indeed have p-value plots that are close to the uniform distribution under null model (Figure 11 in the same appendix highlights four examples). Due to the idealized null hypothesis, this is strong evidence that these competitions are free from overfitting.

## 3.4 Aggregate view of the accuracy competitions

The previous subsections provided tools for analyzing individual competitions and then relied on a qualitative survey of all accuracy competitions. Due to the many nuances and failure modes in machine learning competitions, we believe that this case-by-case approach is the most suitable for the Kaggle data. But since this approach also carries the risk of missing overall trends in the data, we complement it here with a quantitative analysis of all accuracy competitions.

In order to compare the amount of overfitting across competitions, we compute the *mean accuracy difference* across all submissions in a competition. Specifically, let $\mathcal{C}$ be a set of submissions for a given competition. Then the mean accuracy difference of the competition is defined as

$$\text{mean\_accuracy\_difference} = \frac{1}{|\mathcal{C}|} \sum_{i \in \mathcal{C}} \text{public\_accuracy}(i) - \text{private\_accuracy}(i) \,. \qquad (2)$$

A larger mean accuracy difference indicates more potential overfitting. Note that since accuracy is in a sense the opposite of loss, our computation of mean accuracy difference is the opposite of our earlier expression for the generalization gap.

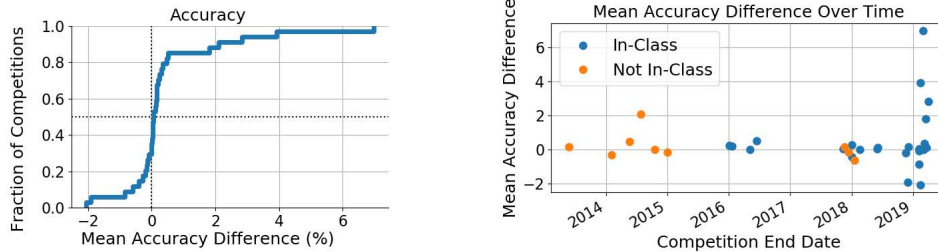

Figure 4: Left: Empirical CDF of the mean accuracy differences (%) for 34 accuracy competitions with at least 1,000 submissions. Right: Mean accuracy differences versus competition end date for the same competitions.

Figure 4 shows the empirical CDF of the mean accuracy differences for all accuracy competitions. While the score differences between $-2\%$ to $+2\%$ are approximately symmetric and centered at zero (as a central limit theorem argument would suggest), the plot also shows a tail with larger score differences consisting of five competitions. Figure 8 in Appendix B.2 aggregates our three types of analysis plots for these competitions.

We have already mentioned three of these outliers in the discussion so far. The worst outlier is Competition 12598 which contains a non-i.i.d. data split (see Section 3.1). Section 3.2 noted that Competitions 3641 and 12681 had very small test sets. Similarly, the other two outliers (#12349 and #5903) have small private test sets of size 209 and 100, respectively. Moreover, the public - private accuracy gap *decreases* among the very best submissions in these competitions (see Figure 8 in Appendix B.2). Hence we do not consider these competitions as examples of adaptive overfitting.

As a second aggregate comparison, the right plot in Figure 4 shows the mean accuracy differences vs. competition end date and separates two types of competitions: in-class and other, which are mainly the "featured" competitions on Kaggle with prize money. Interestingly, many of the competitions with large public - private deviations are from 2019. Moreover, the large deviations are almost exclusively in-class competitions. This may indicate that in-class competitions undergo less quality control than the featured competitions (e.g., have smaller test sets), and that the quality control standards on Kaggle may change over time.

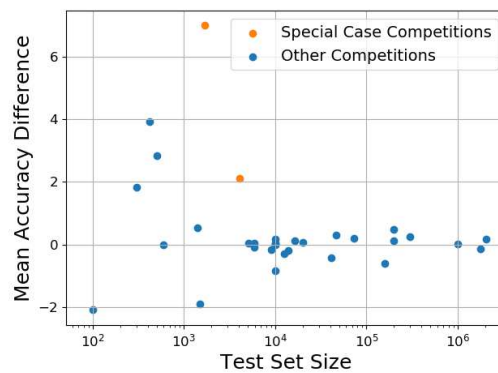

Figure 5: Mean accuracy differences versus test set size (public and private combined) for 32 accuracy competitions with at least 1,000 submissions and available test set size (the test set sizes for two competitions with at least 1,000 submissions were not available from the MetaKaggle dataset).

As a third aggregate comparison, Figure 5 shows the mean accuracy differences vs. test set sizes, with orange dots to indicate competitions we flagged as having either a known or a non-i.i.d. public vs. private test set split. Although a reliable recommendation for applied machine learning will require broader investigation, our results for accuracy competitions suggest that at least 10,000 examples is a reasonable minimum test set size to protect against adaptive overfitting.

### 3.5 Did we observe overfitting?

The preceding subsections provided an increasingly fine-grained analysis of the Kaggle competitions evaluated with classification accuracy. The scatter plots for all submissions in Section 3.1 show a good fit to the $y = x$ diagonal with small and approximately symmetric deviations from the diagonal. This is strong evidence that the overall competition is not affected by substantial overfitting. When restricting the plots to the top submissions in Section 3.2, the picture becomes more varied but the largest overfitting effect size (public - private accuracy deviation) is still small. Thus for some competitions we observe evidence of mild overfitting, but always with small effect size. In both sections we have identified outlier competitions that do not follow the overall trend but also have issues such as small test sets or non-i.i.d. splits. At the finest level of our analysis (Section 3.3), the p-value plots show that the data is only sometimes in agreement with an idealized null model for no overfitting.

Overall we see more signs of overfitting as we sharpen our analysis to highlight smaller effect sizes. So while we cannot rule out every form of overfitting, we view our findings as evidence that overfitting did not pose a significant danger in the most popular classification accuracy competitions on Kaggle. In spite of up to 35,000 submissions to these competitions, there are no large overfitting effects. So while overfitting may have occurred to a small extent, it did not invalidate the overall conclusions from the competitions such as which submissions rank among the top or how well they perform on the private test split.

## 4 Classification competitions with further evaluation metrics

In addition to classification accuracy competitions, we also surveyed competitions evaluated with AUC, MAP@K, LogLoss, and MulticlassLoss. Unfortunately, the Meta Kaggle dataset contains only *aggregate* scores for the public and private test set splits, not the loss values for individual predictions. For the accuracy metric, aggregate scores are sufficient to compute statistical measures of uncertainty such as the error bars in the scatter plots (Sections 3.1 & 3.2) or the p-value plots (Section 3.3). However, the aggregate data is insufficient to compute similar quantities for the other classification metrics. For instance, the lack of example-level scores precludes the use of standard tools such as the bootstrap or permutation tests, as we are unable to re-sample the test set. Hence our analysis here is more qualitative than for accuracy competitions in the preceding section. Nevertheless, inspecting the scatter plots can still convey overall trends in the data.

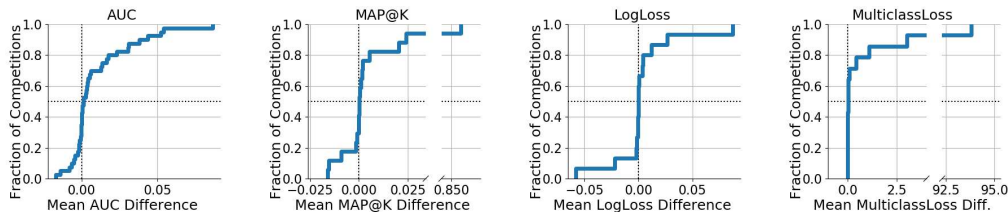

Figure 6: Empirical CDF of mean score differences (across all pre-deadline submissions) for 40 AUC competitions, 17 MAP@K competitions, 15 LogLoss competitions, and 14 MulticlassLoss competitions.

Appendices C to F contain the scatter plots for all submissions and for the top 10% of submissions to these competitions. The overall picture is similar to the accuracy plots: many competitions have scatter plots with a good linear fit close to the $y = x$ diagonal. There is more variation in the top 10% plots but due to the lack of error bars it is difficult to attribute this to overfitting vs. random noise. As before, a small number of competitions show more variation that may be indicative of overfitting. In all cases, the Kaggle website (data descriptions and discussion forums) give possible reasons for this behavior (non-i.i.d. splits, competitions with two stages and different test sets, etc.). Thus, we view these competitions as outliers that do not contradict the overall trend.

Figure 6 shows plots with aggregate statistics (mean score difference) similar to Section 3.4. As for the accuracy competitions, the empirical distribution has an approximately symmetric part centered at 0 (no score change) and a tail with larger score differences that consists mainly of outlier competitions.

# 5    Related work

As mentioned in the introduction, the reproducibility experiment of Recht et al. [16] also points towards a surprising absence of adaptive overfitting in popular machine learning benchmarks. However, there are two important differences to our work. First, Recht et al. [16] assembled new test sets from scratch, which makes it hard to disentangle the effects of adaptive overfitting and distribution shifts. In contrast, most of the public / private splits in Kaggle competitions are i.i.d., which removes distribution shifts as a confounder. Second, Recht et al. [16] only investigate two image classification benchmarks on which most models come from the same model class (CNNs) [9, 11]. We survey 120 competitions on which the Kaggle competitors experimented with a broad range of models and training approaches. Hence our conclusions about overfitting apply to machine learning more broadly.

The adaptive data analysis literature [6, 17] provides a range of theoretical explanations for how the common machine learning workflow may implicitly mitigate overfitting [3, 8, 12, 23]. Our work is complementary to these papers and conducts a purely empirical study of overfitting in machine learning competitions. We hope that our findings can help test and refine the theoretical understanding of overfitting in future work.

The Kaggle community has analyzed competition "shake-up", i.e., rank changes between the public and private leaderboards of a competition. We refer the reader to the comprehensive data analysis conducted by Trotman [21] as a concrete example. Focusing on rank changes is complementary to our approach based on submission scores. From a competition perspective where the winning submissions are defined by the private leaderboard rank, large rank changes are indeed undesirable. However, from the perspective of adaptive overfitting, large rank changes can be a natural consequence of random noise in the evaluation. For instance, consider a setting where a large number of competitors submits solutions with very similar public leaderboard scores. Due to the limited size of the public and private test sets, the public scores are only approximately equal to the private scores (even in the absence of any adaptive overfitting), which can lead to substantial rank changes even though all score deviations are small and of (roughly) equal size. Since the ranking approach can result in such "false positives" from the perspective of adaptive overfitting, we decided not to investigate rank changes in our paper. Nevertheless, we note that we also manually compared the shake-up results to our analysis and generally found agreement among the set of problematic competitions (e.g., competitions with non-i.i.d. splits or known public / private splits).

# 6    Conclusion and future work

We surveyed 120 competitions on Kaggle covering a wide range of classification tasks but found little to no signs of adaptive overfitting. Our results cast doubt on the standard narrative that adaptive overfitting is a significant danger in the common machine learning workflow.

Moreover, our findings call into question whether common practices such as limiting test set re-use increase the reliability of machine learning. We have seen multiple competitions where a non-i.i.d. split lead to substantial gaps between public and private scores, suggesting that distribution shifts [7, 15, 16, 20] may be a more pressing problem than test set re-use in current machine learning.

There are multiple directions for empirically understanding overfitting in more detail. Our analysis here focused on classification competitions, but Kaggle also hosts many regression competitions. Is there more adaptive overfitting in regression? Answering this questions will likely require access to the individual predictions of the Kaggle submissions to appropriately handle outlier submissions. In addition, there are still questions among the classification competitions. For instance, one refinement of our analysis here is to obtain statistical measures of uncertainty for competitions evaluated with metrics such as AUC (which will also require a more fine-grained version of the Kaggle data). Finally, another important question is whether other competition platforms such as CodaLab [1] or EvalAI [22] also show little signs of adaptive overfitting.

**Acknowledgments**

This research was generously supported in part by ONR awards N00014-17-1-2191, N00014-17-1-2401, and N00014-18-1-2833, the DARPA Assured Autonomy (FA8750-18-C-0101) and Lagrange (W911NF-16-1-0552) programs, a Siemens Futuremakers Fellowship, and an Amazon AWS AI Research Award.

## Footnotes

[2] `https://www.kaggle.com/kaggle/meta-kaggle`

[3] Since test examples without labels are available, contestants may still overfit to $S_{\text{private}}$ using an unsupervised approach. While such overfitting may have occurred in a limited number of competitions, we base our analysis on the assumption that unsupervised overfitting did not occur widely with effect sizes that are large compared to overfitting to the public test split.

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
