[Supplementary Material]

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

# A  Overview of Kaggle competitions and our resulting selection criteria

At the time of writing, the MetaKaggle dataset has 1,461 competitions. Due to the variety of different scoring mechanisms used on Kaggle and intricacies of individual competitions, it is challenging to analyze the entire datast in a coherent way. To limit the scope of our investigation and allow for a detailed analysis, we restricted our attention to a subset of competitions based on the following two criteria.

Figure 7: Overview of the Kaggle competitions. The left plot shows the distribution of submissions per competition. The right plot shows the score types that are most common among the competitions with at least 1,000 submissions.

**Number of submissions.**  Kaggle competitions have a wide range of submissions, ranging from 0 for the bottom 10% to 3,159 for the most popular 10% of competitions (and 132,097 for the most popular competition). The left plot in Figure 7 shows the distribution of competition sizes. From the perspective of adaptive overfitting, competitions with many submissions are more interesting since a larger number of submissions indicate more potential adaptivity. Moreover, overfitting on the most popular competitions would also be more important to understand as it would undermine the credibility of the competition paradigm. So to limit the number of competitions, we included only competitions with at least 1,000 submissions before the corresponding competition deadline. This left a total of 262 competitions.

**Score type.**  Depending on the competition, the performance of a submissions is measured with one of various scoring mechanisms. The list of scoring mechanisms involves well-known score types (accuracy, AUC, RMSE) and a long tail of more specialized metrics (see the right plot in Figure 7. We focus on score types used for classification tasks with at least 10 competitions (passing our submission count filter) so we can compare competitions with multiple other competitions using the same score type. This avoids difficulties such as comparing classification vs. regression competitions or bounded vs. unbounded scores. Moreover, the most popular score types are also the most relevant for measuring the amount of overfitting in machine learning competitions. The classification score types with at least 10 competitions are classification accuracy, AUC (area under curve), MAP@K (mean average precision), the logistic loss, and a multiclass variant of the logistic loss.

In total, the above exclusion rules leave 120 competitions. It would be possible to restrict the set of competitions even further, e.g., by only including "featured" competitions (the competitions promoted by Kaggle, usually with prize money) and excluding other competition types such as "in class" competitions which are mainly used for educational purposes. Another possible filter would be to exclude competitions with only small test set sizes since the resulting scores are inherently more noisy. However, we decided to minimize the exclusion rules so we can analyze a large number of competitions and possibly find effects of features such as competition type and test set size.

# B  Accuracy

## B.1  Accuracy: competition info

Table 2: Competitions scored with accuracy with greater than 1000 submissions. $n_{\text{public}}$ is the size of the public test set and $n_{\text{private}}$ is the size of the private test set. N/A means that we could not access the competition data to compute the dataset sizes. A * after the competition name means the name was slightly edited to fit into the table.

| | Accuracy | | | |
|---|---|---|---|---|
| ID | Name | # Sub. | $n_{\text{public}}$ | $n_{\text{private}}$ |
| 3362 | Dogs vs. Cats | 1,225 | 3,750 | 8,751 |
| 3366 | The Black Box Learning Challenge* | 1,924 | 5,000 | 5,000 |
| 3428 | Data Science London + Scikit-learn | 1,477 | 2,700 | 6,299 |
| 3641 | DecMeg2014 - Decoding the Human Brain | 4,507 | 1,745 | 2,313 |
| 3649 | CIFAR-10 - Object Recognition in Images | 1,634 | 300 | 9,700 |
| 3788 | Allstate Purchase Prediction Challenge | 24,532 | 59,657 | 139,199 |
| 4554 | MSU Visits* | 1,288 | 30,000 | 270,000 |
| 4821 | 104-1 MLDS_Final | 1,060 | 36,400 | 36,401 |
| 4857 | Let's Overfit | 1,080 | 500,000 | 500,000 |
| 5275 | Can we predict voting outcomes? | 35,247 | 696 | 696 |
| 5896 | Prediction Reviews Sentiment Analysis* | 1,770 | 50 | 50 |
| 5903 | Prediction Reviews Sentiment Analysis (Light)* | 3,528 | 400 | 100 |
| 5942 | Identify Me If You Can – Yandex & MIPT | 1,242 | 20,588 | 20,589 |
| 6109 | Identify Me If You Can | 1,819 | 23,236 | 23,237 |
| 7042 | Text Normalization Challenge - English Language | 1,834 | 20,446 | 2,024,166 |
| 7115 | Cdiscount's Image Classification Challenge | 5,859 | 530,455 | 1,237,727 |
| 7563 | Cover Type Prediction of Forests | 3,077 | 6,000 | 14,000 |
| 7634 | TensorFlow Speech Recognition Challenge | 24,263 | 3,171 | 155,365 |
| 8307 | Data Science Nigeria Telecoms Churn | 1,259 | 180 | 420 |
| 9051 | Digit Recongnizer 1438 | 1,640 | 2,576 | 2,575 |
| 9491 | ML-2018spring-hw5 | 2,247 | 100,000 | 100,000 |
| 10851 | CSE158 fa18 Category Prediction | 3,872 | 7,000 | 7,000 |
| 11842 | AIA image classification by CNN | 3,415 | 1,350 | 150 |
| 12187 | CS 4740 Project 4 Revised | 1,086 | N/A | N/A |
| 12349 | 2019 1st ML month with KaKR | 3,307 | 209 | 209 |
| 12357 | UC Berkeley CS189 HW1 (MNIST) | 1,480 | 3,000 | 7,000 |
| 12358 | UC Berkeley CS189 HW1 (SPAM) | 1,527 | 1,757 | 4,100 |
| 12359 | UC Berkeley CS189 HW1 (CIFAR-10) | 1,336 | 3,000 | 7,000 |
| 12598 | TAU Robot Surface Detection | 1,574 | 852 | 853 |
| 12681 | PadhAI: Text - Non Text Classification Level 2 | 1,185 | 90 | 210 |
| 13034 | UC Berkeley CS189 HW3 (MNIST) | 1,193 | 3,000 | 7,000 |
| 13036 | UC Berkeley CS189 HW3 (SPAM) | 1,080 | 1,757 | 4,100 |
| 13078 | Homework 2 Part 2 - Classification | 2,245 | N/A | N/A |
| 13383 | ML2019spring-hw2 | 2,802 | 8,141 | 8,141 |

## B.2 Accuracy: outlier competitions

Figure 8 shows the plots corresponding to the three levels of overfitting analysis for the mean accuracy difference outliers discussed in Section 3.4.

By reading the Kaggle forums and the description of the dataset construction, we were able to determine a possible cause of overfitting for most of the outlier competitions. In some cases, the competition hosts did not create the public and private test splits in a truly i.i.d. manner, and competitors were able to exploit the difference in distribution. In other cases, the size of the test set was extremely small. The list of outlier competitions and corresponding cause of overfitting follow. We also include links to relevant descriptions of the data or discussions on the Kaggle forums.

- **Competition 3641: DecMeg2014 - Decoding the Human Brain** The test set contains data generated from 7 subjects and the public and private splits were created in a non-i.i.d. manner using data from individual subjects. `https://www.kaggle.com/c/decoding-the-human-brain/data`

- **Competition 5903: Prediction Reviews Sentiment Analysis (Light)** The competition uses a small test set consisting of 500 examples (400 in the public test set and 100 in the private test set).

- **Competition 12349: 2019 1st ML month with KaKR** The competition uses a small test set consisting of 418 examples (209 in the private test set and 209 in the public test set.)

- **Competition 12598: TAU Robot Surface Detection** The dataset is a time series that consists of measurements of acceleration, velocity and orientation. In some of the measurement locations, the floor has a slope that makes the orientation channels informative (though the orientation terms are designed to not be informative for the particular task). In order to mitigate this effect, the organizers split the time series data into shorter subsequences and assigned separate subsequences to the public and private split. Thus, the public and private data were not created from a truly i.i.d. split. [4]

- **Competition 12681: PadhAI: Text - Non Text Classification Level 2** The competition uses a small test set consisting of 300 examples (90 in the public test set and 210 in the private test set).

Figure 8: The three main plots for the outliers in mean score difference discussed in Section 3.4: private versus public accuracy for all submissions (left), the top 10% of submissions ranked by public accuracy (center), and the empirical CDFs of p-Values (right). For the scatter plots, each point corresponds to an individual submission (shown with 95% Clopper-Pearson confidence intervals). The right figure includes empirical CDFs for an idealized null model that assumes no overfitting (see Section 3.3).

## B.3 Accuracy: all submissions

Figure 9: Private versus public accuracy for all submissions for the most popular Kaggle accuracy competitions. Each point corresponds to an individual submission (shown with 95% Clopper-Pearson confidence intervals).

Figure 9: Private versus public accuracy for all submissions for the most popular Kaggle accuracy competitions. Each point corresponds to an individual submission (shown with 95% Clopper-Pearson confidence intervals).

## B.4 Accuracy: top 10% of submissions

Figure 10: Private versus public accuracy for top 10% of submissions for the most popular Kaggle accuracy competitions. Each point corresponds to an individual submission (shown with 95% Clopper-Pearson confidence intervals).

Figure 10: Private versus public accuracy for all submissions for the most popular Kaggle accuracy competitions. Each point corresponds to an individual submission (shown with 95% Clopper-Pearson confidence intervals).

## B.5 Accuracy: p-values

### B.5.1 Computation of P-Values

For a given submission, the public and private test accuracies will rarely exactly match. We would like to understand exactly how much of this discrepancy can be attributed to random variation as opposed to overfitting. Using only MetaKaggle data, we compute *exact* p-values for a simple null mode.

Fix a classifier that correctly classifies $m$ points out of a total test set of size $n_{\text{total}}$ and incorrectly classifies the remaining $n_{\text{total}} - m$ points. We randomly split these $n_{\text{total}}$ points into public and private test sets of size $n_{\text{public}}$ and $n_{\text{private}}$, respectively, with $n_{\text{public}} + n_{\text{private}} = n_{\text{total}}$. Under the null model, every partition of the points is equally likely. Dividing points in this fashion leads to $m_{\text{public}}$ and $m_{\text{private}}$ successes on each subset. Let $\Delta$ be the observed difference in public and private accuracies. Then, we're interested in computing

$$\texttt{p\_value} = \Pr\left\{\left|\frac{m_{\text{private}}}{n_{\text{private}}} - \frac{m_{\text{public}}}{n_{\text{public}}}\right| > \Delta\right\}. \tag{3}$$

Fixing $m_{\text{private}}$ also determines $m_{\text{public}}$, and for any fixed $m_{\text{private}} = k$, the probability of sampling a private test set of with $k$ correctly classified points is

$$\frac{\binom{m}{k}\binom{n-m}{n_{\text{private}}-k}}{\binom{n}{n_{\text{private}}}}. \tag{4}$$

Therefore, we can compute the p-value in equation (3) as

$$\Pr\left\{\left|\frac{m_{\text{private}}}{n_{\text{private}}} - \frac{m_{\text{public}}}{n_{\text{public}}}\right| > \Delta\right\} = \sum_{k\in[m]:\left|\frac{k}{n_{\text{private}}} - \frac{(m-k)}{n_{\text{public}}}\right|>\Delta} \frac{\binom{m}{k}\binom{n-m}{n_{\text{private}}-k}}{\binom{n}{n_{\text{private}}}}, \tag{5}$$

where $[m] = \{0, 1, 2, \ldots, m\}$. Evaluating this summation only requires access to $n_{\text{total}}, n_{\text{public}}, n_{\text{private}}$, and $\Delta$, all of which are available in the MetaKaggle dataset. Hence, a p-value can be readily evaluated for each submission. Computing ratios of binomial coefficients like (4) is numerically unstable and computationally expensive for large $n$. Therefore, in our implementation, we work in log-space and use Stirling's approximation for the binomial coefficients to efficiently and reliably approximate (5).

### B.5.2 Accuracy: p-value plots

Figure 11: Empirical CDFs for an idealized null model that assumes no overfitting (full details in Section 3.3).

Figure 12: Empirical CDFs for an idealized null model that assumes no overfitting (full details in Section 3.3)

Figure 12: Empirical CDFs for an idealized null model that assumes no overfitting (full details in Section 3.3)

# C AUC

## C.1 AUC: competition info

Table 3: Competitions scored with AUC with greater than 1000 submissions. $n_{\text{public}}$ is the size of the public test set and $n_{\text{private}}$ is the size of the private test set. N/A means that we could not access the competition data to compute the dataset sizes. A * after the competition name means the name was slightly edited to fit into the table.

| | AUC | | | |
|---|---|---|---|---|
| ID | Name | # Sub. | $n_{\text{public}}$ | $n_{\text{private}}$ |
| 2439 | INFORMS Data Mining Contest 2010 | 1,483 | 254 | 2,285 |
| 2445 | Predict Grant Applications | 2,800 | 544 | 1,632 |
| 2464 | IJCNN Social Network Challenge | 1,124 | 1,792 | 7,168 |
| 2478 | Stay Alert! The Ford Challenge | 1,402 | 36,252 | 84,588 |
| 2489 | Don't Overfit! | 3,775 | 1,975 | 17,775 |
| 2551 | Give Me Some Credit | 7,724 | 30,451 | 71,052 |
| 3338 | Amazon.com - Employee Access Challenge | 16,896 | 17,676 | 41,245 |
| 3353 | The Marinexplore and Cornell Univ. Whale Detection* | 3,295 | 16,351 | 38,152 |
| 3469 | Influencers in Social Networks | 2,109 | 2,976 | 2,976 |
| 3509 | The ICML 2013 Whale Challenge - Right Whale Redux | 1,005 | 7,641 | 17,828 |
| 3524 | Accelerometer Biometric Competition | 7,130 | 8,102,160 | 18,905,040 |
| 3526 | StumbleUpon Evergreen Classification Challenge | 7,509 | 634 | 2,537 |
| 3774 | CONNECTOMICS | 1,458 | 75,742 | 75,743 |
| 3897 | Acquire Valued Shoppers Challenge | 25,205 | N/A | N/A |
| 3926 | Predicting Excitement at DonorsChoose.org* | 12,530 | 11,193 | 33,579 |
| 3933 | MLSP 2014 Schizophrenia Classification Challenge | 2,246 | 0 | 119,749 |
| 3960 | American Epilepsy Society Seizure Prediction Challenge | 17,782 | 1,574 | 2,361 |
| 4031 | Driver Telematics Analysis | 36,072 | N/A | N/A |
| 4043 | BCI Challenge @ NER 2015 | 4,348 | 680 | 2,721 |
| 4294 | Facebook Recruiting IV: Human or Robot? | 13,559 | 1,410 | 3,290 |
| 4366 | West Nile Virus Prediction | 29,963 | 2,326 | 113,967 |
| 4487 | Springleaf Marketing Response | 39,439 | 43,570 | 101,662 |
| 4493 | Truly Native? | 3,224 | 1,360 | 2,040 |
| 4657 | Homesite Quote Conversion | 36,387 | 52,151 | 121,685 |
| 4986 | Santander Customer Satisfaction | 93,584 | 37,909 | 37,909 |
| 5167 | PRED 411-2016_04-U2-INSURANCE-A* | 1,436 | 1,070 | 1,071 |
| 5174 | Avito Duplicate Ads Detection | 8,157 | 657,602 | 657,603 |
| 5261 | Predicting Red Hat Business Value | 33,696 | 149,606 | 349,081 |
| 5390 | Melbourne Univ. Seizure Prediction* | 10,083 | N/A | N/A |
| 6242 | Catch Me If You Can: Intruder Detection* | 2,479 | 41,398 | 41,399 |
| 7162 | WSDM - KKBox's Music Recommendation Challenge | 15,555 | 1,278,396 | 1,278,395 |
| 8227 | 2018 Spring CSE6250 HW1 | 1,170 | 244,173 | 244,173 |
| 8540 | TalkingData AdTracking Fraud Detection Challenge | 68,594 | 3,382,284 | 15,408,185 |
| 9120 | Home Credit Default Risk | 132,097 | 9,749 | 38,995 |
| 10683 | Microsoft Malware Prediction | 43,702 | 4,947,549 | 2,905,704 |
| 11803 | BGU - Machine Learning | 2,079 | 6,849 | 5,603 |
| 11848 | Histopathologic Cancer Detection | 20,352 | 28,154 | 29,304 |
| 12512 | 2019 Spring CSE6250 BDH | 1,396 | 244,173 | 244,173 |
| 12558 | WiDS Datathon 2019 | 3,021 | 4,312 | 2,222 |
| 12904 | Caltech CS 155 2019 Part 1 | 1,078 | 1,264,122 | 1,264,122 |

## C.2 AUC: outlier competitions

For AUC, we also investigated competitions whose linear fit was a bad approximation to the diagonal $y = x$ line. We first identified these competitions using visual inspection of the plots in Section C.3.

By reading the Kaggle forums and the description of the dataset construction, we were able to determine a possible cause of overfitting for most of the outlier competitions. In most cases, the competition hosts did not create the public and private test splits in a truly i.i.d. manner, and competitiors were able to exploit the difference in distribution. In other cases, the size of the test set was extremely small. The list of outlier competitions and corresponding cause of overfitting follow. We also include links to relevant descriptions of the data or discussions on the Kaggle forums.

- **Competition 3774: CONNECTOMICS** Participants knew which test data belonged in the public and private splits because the splits were provided in separate files. `https://www.kaggle.com/c/connectomics/data`

- **Competition 3926: KDD Cup 2014 - Predicting Excitement at DonorsChoose.org**: Public and private splits are not i.i.d. and likely were created by separating donations projects by the date they were proposed. `https://www.kaggle.com/c/kdd-cup-2014-predicting-excitement-at-donors-choose/discussion/9772#latest-50827`

- **Competition 3933: MLSP 2014 Schizophrenia Classification Challenge** The test set contains data generated from 58 subjects and the public and private splits were created in a non-i.i.d. manner using data from individual subjects. `https://www.kaggle.com/c/mlsp-2014-mri/discussion/10135#latest-54483`

- **Competition 4043: BCI Challenge @ NER 2015**: The test set contains data generated from 10 subjects and the public and private splits were created in a non-i.i.d. manner using data from individual subjects `https://www.kaggle.com/c/inria-bci-challenge/discussion/12613#latest-65652`.

- **Competition 8540: TalkingData AdTracking Fraud Detection Challenge** A second, larger test set that was unintentionally released at the start of the competition and then participants were permitted but not required to use the data. `https://www.kaggle.com/c/talkingdata-adtracking-fraud-detection/discussion/52658#latest-315882`

- **Competition 10683: Microsoft Malware Prediction** The private test data included several severe outliers not present in the public test data, indicating that the public and private split are not i.i.d. `https://www.kaggle.com/c/microsoft-malware-prediction/discussion/84745`

## C.3 AUC: all submissions

Figure 13 shows the private versus public AUC scatter plots for each AUC competition with over 1000 submissions in the MetaKaggle dataset.

Figure 13: Private versus public AUC for all submissions for Kaggle AUC competitions.

Figure 13: Private versus public AUC for all submissions for Kaggle AUC competitions.

## C.4 AUC: top 10% of submissions

**Competition 2439**
Top 10% (n=148)

**Competition 2445**
Top 10% (n=280)

**Competition 2464**
Top 10% (n=112)

**Competition 2489**
Top 10% (n=377)

**Competition 2478**
Top 10% (n=140)

**Competition 2551**
Top 10% (n=772)

**Competition 3338**
Top 10% (n=1689)

**Competition 3353**
Top 10% (n=329)

**Competition 3469**
Top 10% (n=210)

**Competition 3509**
Top 10% (n=100)

**Competition 3524**
Top 10% (n=713)

**Competition 3526**
Top 10% (n=750)

**Competition 3774**
Top 10% (n=145)

**Competition 3897**
Top 10% (n=2520)

**Competition 3926**
Top 10% (n=1253)

**Competition 3933**
Top 10% (n=224)

**Competition 3960**
Top 10% (n=1778)

**Competition 4031**
Top 10% (n=3607)

**Competition 4043**
Top 10% (n=434)

**Competition 4294**
Top 10% (n=1355)

**Competition 4366**
Top 10% (n=2996)

**Competition 4487**
Top 10% (n=3943)

**Competition 4493**
Top 10% (n=322)

**Competition 4657**
Top 10% (n=3638)

**Competition 4986**
Top 10% (n=9358)

**Competition 5167**
Top 10% (n=143)

**Competition 5174**
Top 10% (n=815)

**Competition 5261**
Top 10% (n=3369)

● Submission ── Linear fit ┅ y=x

Figure 14: Private versus public AUC for top 10% of submissions for Kaggle AUC competitions.

# D   MAP@K

## D.1   MAP@K: competition info

Our MAP@K (Mean Average Precision @ K) category includes competitions marked in the MetaKaggle dataset as MAP@{K}, MAP@k, MAP@3, AP@{K}, and MAP@{K}_OLD, as these metrics are functionally equivalent. MAP@K is typically used for tasks that require predicting a ranked list of items; for instance, listing $K$ advertisements in order based on how likely a particular user is to click on them. In this example, the 'Mean' in 'Mean Average Precision' is taken over all users. For the $i$'th user, 'Average Precision' of the $K$ predicted advertisements is defined as:

$$\frac{1}{\min(m_i, K)} \sum_{k=1}^{K} P_i(k) I_i(k),$$

where $m_i$ is the number of advertisements that user $i$ actually clicked on, $P_i(k)$ is the precision of the first $k$ recommendations for user $i$ (the fraction of the first $k$ recommended advertisements that the user actually clicked on), and $I_i(k)$ is an indicator variable representing whether the $k$'th advertisement was clicked on. In the MetaKaggle dataset, MAP@{K} and MAP@k denote precisely this Mean Average Precision @ K scoring metric, for varying values of $K$. MAP@3 is the same, with $K = 3$. AP@{K} denotes Average Precision @ K, which is functionally equivalent to MAP@K because such competitions require predicting a single list (and the mean of a single element is itself). MAP@{K}_OLD is also effectively the same metric, except with no pre-specified limit on $K$. In our earlier analogy, this would correspond to recommending all the available advertisements (rather than only the top $K$), so the scoring metric is based solely on the ordering of the recommendations.

Table 4: Competitions scored with MAP@K with greater than 1000 submissions. $n_{\text{public}}$ is the size of the public test set and $n_{\text{private}}$ is the size of the private test set. A * after the competition name means the name was slightly edited to fit into the table. A ** indicates competitions where the public test set and the private test set are identical and both use all of the test data.

| MAP@K | | | | |
|---|---|---|---|---|
| ID | Name | # Sub. | $n_{\text{public}}$ | $n_{\text{private}}$ |
| 2748 | KDD Cup 2012, Track 1 | 13,077 | 710,267 | 629,860 |
| 2947 | Facebook Recruiting Competition | 3,554 | 65,647 | 196,941 |
| 3288 | Event Recommendation Engine Challenge | 3,021 | 10,238 | 10,238** |
| 3445 | KDD Cup 2013 (Track 1)* | 9,426 | 2,244 | 2,244** |
| 3929 | The Hunt for Prohibited Content | 5,001 | 675,622 | 675,621 |
| 4481 | Coupon Purchase Prediction | 18,487 | 6,870 | 16,030 |
| 5056 | Expedia Hotel Recommendations | 22,713 | 834,321 | 1,693,923 |
| 5186 | Facebook V: Predicting Check Ins | 15,127 | 4,445,370 | 4,445,369 |
| 5497 | Outbrain Click Prediction | 6,654 | 9,667,549 | 22,557,613 |
| 5558 | Santander Product Recommendation | 28,773 | 278,884 | 650,731 |
| 6818 | Humpback Whale Identification | 37,529 | 1,592 | 6,368 |
| 7666 | CS5785 Fall 2017 Final Exam | 3,898 | 1,000 | 1,000 |
| 8396 | Google Landmark Retrieval Challenge | 3,123 | 40,019 | 77,685 |
| 8857 | Recipient prediction 2018 | 1,154 | 1,000 | 1,000 |
| 8900 | Freesound Audio Tagging Challenge* | 5,684 | 282 | 9,118 |
| 10200 | Quick, Draw! Doodle Recognition Challenge | 21,407 | 10,098 | 102,101 |
| 12088 | CS5785 Fall 2018 Final | 1,877 | 1,000 | 1,000 |

## D.2   MAP@K: outlier competitions

For MAP@K, we also investigated competitions whose linear fit was a bad approximation to the diagonal $y = x$ line. We first identified these competitions using visual inspection of the plots in Section D.3.

By reading the Kaggle forums and the description of the dataset construction, we were able to determine a possible cause of overfitting for most of the outlier competitions. In most cases, the

public and private test set splits were known to the participants. We also include links to relevant descriptions of the data or discussions on the Kaggle forums.

- **Competition 2748: KDD Cup 2012, Track 1** Participants knew which test data belonged in the public and private splits because the splits were based on timestamp. `https://www.kaggle.com/c/kddcup2012-track1/discussion/1632#latest-9759`

- **Competition 3288: Event Recommendation Engine Challenge** The public and private test sets were released in separate stages, so in the final stage of the competition participants knew which test data belonged in each split. `https://www.kaggle.com/c/event-recommendation-engine-challenge/discussion/3840#latest-20556`

- **Competition 3445: KDD Cup 2013 - Author-Paper Identification Challenge (Track 1)** Participants knew which test data belonged in the public and private splits because the splits were released at different stages of the competition. `https://www.kaggle.com/c/kdd-cup-2013-author-paper-identification-challenge/overview/timeline`

## D.3 MAP@K: all submissions

Figure 15: Private versus public MAP@K for all submissions for Kaggle MAP@K competitions.

## D.4    MAP@K: top 10% of submissions

Figure 16: Private versus public MAP@K for top 10% of submissions for Kaggle MAP@K competitions.

# E  MulticlassLoss

Our MulticlassLoss category includes competitions marked in the MetaKaggle dataset as Multiclass-Loss and MulticlassLossOld, as these metrics are functionally equivalent.

## E.1  MulticlassLoss: competition info

Table 5: Competitions scored with MulticlassLoss with greater than 1000 submissions. $n_{\text{public}}$ is the size of the public test set and $n_{\text{private}}$ is the size of the private test set.

| MulticlassLoss | | | | |
|---|---|---|---|---|
| ID | Name | # Sub. | $n_{\text{public}}$ | $n_{\text{private}}$ |
| 3043 | Predict Closed Questions on Stack Overflow | 1,431 | 0 | 81,686 |
| 3978 | National Data Science Bowl | 15,121 | 9,128 | 121,273 |
| 4117 | Microsoft Malware Classification Challenge (BIG 2015) | 7,670 | 3,262 | 7,611 |
| 4280 | Otto Group Product Classification Challenge | 43,524 | 101,058 | 43,310 |
| 4521 | Right Whale Recognition | 4,789 | 2,424 | 4,501 |
| 4588 | Telstra Network Disruptions | 19,603 | 3,240 | 7,931 |
| 4654 | Walmart Recruiting: Trip Type Classification | 13,135 | 28,702 | 66,972 |
| 5048 | State Farm Distracted Driver Detection | 25,591 | 15,148 | 64,579 |
| 5340 | TalkingData Mobile User Demographics | 24,631 | 33,621 | 78,450 |
| 5568 | The Nature Conservancy Fisheries Monitoring | 2,100 | 1,132 | 13,021 |
| 5590 | Two Sigma Connect: Rental Listing Inquiries | 47,044 | 29,864 | 44,796 |
| 6243 | Intel & MobileODT Cervical Cancer Screening | 1,367 | 589 | 3,943 |
| 6841 | Personalized Medicine: Redefining Cancer Treatment | 3,056 | 2,267 | 3,401 |
| 7516 | Spooky Author Identification | 10,640 | 2,518 | 5,874 |

## E.2  MulticlassLoss: outlier competitions

For MulticlassLoss, we investigated competitions whose linear fit was a bad approximation to the diagonal $y = x$ line. We first identified these competitions using visual inspection of the plots in Section E.3.

By reading the Kaggle forums and the description of the dataset construction, we were able to determine a possible cause of overfitting for most of the outlier competitions. In most cases, the competition hosts did not create the public and private test splits in a truly i.i.d. manner, and competitiors were able to exploit the difference in distribution. In other cases, the size of the test set was extremely small. The list of outlier competitions and corresponding cause of overfitting follow. We also include links to relevant descriptions of the data or discussions on the Kaggle forums.

- **Competition 3043: Predict Closed Questions on Stack Overflow** All test data was considered private, and the public leaderboard was identical to the private leaderboard. `https://www.kaggle.com/c/predict-closed-questions-on-stack-overflow/leaderboard`

- **Competition 5568: The Nature Conservancy Fisheries Monitoring** The test set was released in two stages. `https://www.kaggle.com/c/the-nature-conservancy-fisheries-monitoring/data`

- **Competition 6243: Intel & MobileODT Cervical Cancer Screening** The test set was released in two stages. `https://www.kaggle.com/c/intel-mobileodt-cervical-cancer-screening/data`

- **Competition 6841: Personalized Medicine: Redefining Cancer Treatment** The test set was released in two stages because a team discovered the test data from Stage 1, including labels, on a public website. `https://www.kaggle.com/c/msk-redefining-cancer-treatment/discussion/36383#latest-223033`

## E.3 MulticlassLoss: all submissions

Figure 17: Private versus public MulticlassLoss for all submissions for Kaggle MulticlassLoss competitions.

## E.4 MulticlassLoss: top 10% of submissions

Figure 18: Private versus public MulticlassLoss for top 10% of submissions for Kaggle MulticlassLoss competitions.

# F LogLoss

## F.1 LogLoss: competition info

Table 6: Competitions scored with LogLoss with greater than 1000 submissions. $n_{\text{public}}$ is the size of the public test set and $n_{\text{private}}$ is the size of the private test set. N/A means that we could not access the competition data to compute the dataset sizes. A * after the competition name means the name was slightly edited to fit into the table.

| | **LogLoss** | | | |
|---|---|---|---|---|
| ID | Name | # Sub. | $n_{\text{public}}$ | $n_{\text{private}}$ |
| 2780 | Predicting a Biological Response | 8,843 | 625 | 1,876 |
| 2984 | Practice Fusion Diabetes Classification | 2,200 | 1,245 | 3,734 |
| 3377 | ICDAR2013 - Gender Prediction from Handwriting | 1,884 | 286 | 486 |
| 3934 | Display Advertising Challenge | 8,640 | 1,208,427 | 4,833,708 |
| 3984 | Tradeshift Text Classification | 5,659 | 163,525 | 381,557 |
| 4120 | Click-Through Rate Prediction | 31,019 | 915,493 | 3,661,972 |
| 4438 | Avito Context Ad Clicks | 5,951 | 4,788,454 | 11,173,061 |
| 4852 | BNP Paribas Cardif Claims Management | 54,519 | 37,750 | 76,643 |
| 4862 | March Machine Learning Mania 2016 | 1,054 | 68 | 2,210 |
| 6004 | Data Science Bowl 2017 | 1,694 | N/A | N/A |
| 6277 | Quora Question Pairs | 53,927 | 140,748 | 2,205,048 |
| 7163 | WSDM - KKBox's Churn Prediction Challenge | 6,256 | 361,788 | 542,683 |
| 7380 | Statoil/C-CORE Iceberg Classifier Challenge | 42,936 | 674 | 7,750 |
| 7823 | Attrition de clientes | 1,911 | 15,000 | 15,000 |
| 8310 | Google Cloud & NCAA® ML Competition 2018* | 1,660 | 342 | 11,049 |

## F.2 LogLoss: outlier competitions

For LogLoss, we observed one competition (Competition 6004: Data Science Bowl 2017) whose linear fit was a bad approximation the diagonal $y = x$ line. The competition shows approximately 13 outliers when looking at the scatter plot for the top 10% of submissions (see Figure 20).

The competition is unique in two aspects: first, the competition used two stages with separate test sets released at different times and second, participants were allowed to use external data. Participants were eventually given all the labels to the Stage 1 test data, but the Stage 2 test data was different in distribution from the Stage 1 test data (see `https://www.kaggle.com/c/data-science-bowl-2017/discussion/31383#latest-174302`). It is unclear whether the final private test data was generated from an i.i.d. split with either the Stage 1 test data or the Stage 2 data. Moreover, training the model on external data violates our assumption that the models were only trained and evaluated on i.i.d. data.

## F.3 LogLoss: all submissions

Figure 19: Private versus public LogLoss for all submissions for Kaggle LogLoss competitions.

## F.4    LogLoss: top 10% of submissions

Figure 20: Private versus public LogLoss for top 10% of submissions for Kaggle LogLoss competitions.

# G   Mean score differences over time

Figure 21: Mean score differences (across all pre-deadline submissions) versus competition end date for all classification evaluation metrics with at least 10 competitions with at least 1000 submissions each. Score difference is public minus private for metrics where a higher score is better (AUC, CategorizationAccuracy (included in the main text), and MAP@K) and private minus public for metrics where a lower score is better (LogLoss and MulticlassLoss). Blue dots show InClass competitions and orange dots show all other competition types (Featured, Research, Getting Started, and Playground).