[Reviews · NeurIPS 2019]

Reviewer 1



Update: I thank the authors for the feedback and the additional figure. I would gladly vote for acceptance of the paper. ----------------------------------------------- The authors conducted the first large meta-analysis of overfitting due to test set reuse in the machine learning community. They surveyed a wide range of 112 Kaggle competitions and concluded that there is little evidence of substantial overfitting. I found the question very important to the machine learning community. The investigation is thorough that it covered a broad spectrum of datasets. The statistical methods used here are appropriate. And therefore, the conclusion is trustworthy. Also, the paper is well-written. I have a few suggestions, but overall, I think this paper deserves publication in NeurIPS. 1. For the overall conclusion, I think a more precise one would be: on the one hand, there is strong evidence regarding the existence of overfitting (p-values in Sec. 3.3); on the other hand, the extent of overfitting is mild (Sec. 3.1 and 3.2). 2. For the figures (Fig. 1, 2), there are too many things that it is a bit hard to parse. First, there are too many points. A common solution is to reduce the point size or make them transparent. Second, since the linear trend is apparent, I would suggest removing the linear regression line. Also, it would be nice to lay the points above the y=x line so that we can see them. Besides, the confidence intervals are missing for some plots, e.g., Figure 1. 3. It is important to investigate the covariates that are likely to cause overfitting. The authors mentioned that one reason might be the small test data size. Can the authors quantify this by plotting, e.g., the extent of overfitting against the testing data sizes across all competitions considered? It is also helpful to recommend a minimum testing data size for an ML competition to have mild overfitting. 4. Another aspect that people care about these competitions is that if the ranks of submissions are preserved between the public testing dataset and private testing dataset. Can the authors add some results on this? Minor: 5. Line 146: I am curious about the subtleties. 6. Eq. (2): missing bracket inside summation. 7. One more related work on adaptive data analysis to include is [Russo et al, 2016, Controlling...]

Reviewer 2



This study analyses overfitting in machine learning based on a meta-analysis of over hundred Kaggle competitions that comprise a variety of classification tasks. Kaggle provides a leave-out test data set, which is used to publicly rank competing methods; the competing methods can aim to improve their public ranking by improving their method. This can potentially lead to overfitting. The degree of that overfitting can be evaluated since Kaggle also maintains additional leave-out data sets that are used to provide a final ranking for the competing methods. The results indicate that the degree of overfitting is relatively low, and the authors' interpretation is that this demonstrates the robustness of cross-validation as a method development technique. In terms of the strengths and weaknesses The overall Quality of the work is high. Whereas the idea and implementation are relatively straightforward at a conceptual level, this work contributes new empirical information on overfitting, a fundamental subject in machine learning. If there are shortcomings, the limitation of the study only to classification tasks is one; but expanding the present work to other machine learning tasks could be relatively straightforward; the paper introduces the new concept, and the clarity of presentation does benefit from a well-defined scope on classification tasks. Source code is also provided, further supporting the overall quality (transparency, access) of the work. Clarity. The study design and presentation are clear, and the implementation is rigorous. Sufficient empirical analysis and discussion of related work are provided, and potential extensions are discussed (other competition platforms; other modeling tasks; scoring methods to assess overfitting). Originality. The large-scale analysis of overfitting in machine learning studies, implemented based on public competition platforms, seems to be a new idea. The paper also includes interesting reflections on the types of competitions and data set qualities (money prices, data set size..) and how these are reflected in overfitting. The work does not include substantially new theoretical or methodological ideas; the originality is mainly empirical. Significance. The study lays a groundwork for extended analysis of overfitting of different types of machine learning models. It has implications for better understanding of the fundamentals in the field. The weakness is that the improved understanding does not readily translate to pragmatic recommendations for analysis, besides bringing increased confidence in cross-validation as a model training technique. A paper with a similar title has been presented in ICML2019 workshop (https://sites.google.com/view/icml2019-generalization/schedule). The content is not the same but there seems to be notable overlap, and the ICML2019 paper is not cited in the present submission. It would help if the authors can clarify what are the main new contributions of this submission with respect to the ICML2019 workshop paper.

Reviewer 3



UPDATE: I thank the authors for their feedback, which I have read. I am not inclined to alter my score from an 8, but once again emphasize that this paper is a good one and I hope to see it published. ---------------------------------------- Thanks to the authors for the hard work on this paper. ==Originality== The work is original in that it is the first rigorous study of the MetaKaggle dataset as pertaining to the problem of adaptive overfitting. As the authors point out, some prior work has been done but only on a few of the most popular image datasets. ==Quality== The work is of high quality, and the experiments are well designed. It is unfortunate that the authors did not also perform the analyses on the regression problems, and I hope they intend to publish follow-up work. It was not clear to me why the authors conclude that only the accuracy metric has "enough" data for statistical measures. The reasoning in lines 275-278 is insufficient. Please be more concrete here. What notion of "enough data" are you using? ==Clarity== The paper is clear and well-written. ==Significance== In general, I think this paper will be of high significance as it begins to cast doubt on a common fear (adaptive overfitting). However, one point that kept coming up is that some of the datasets were "outliers" because their public and private test sets were not IID. It is true that this paper is entirely about adaptive overfitting in the IID case, so it makes sense to somewhat put aside non-IID cases, but in my personal experience adaptive overfitting is particularly problematic when the data is not quite IID, but close. For example, competition 3641 has 7 subjects, and the public/private split is done on subject. This is actually reasonable when considering how most ML algorithms are actually applied in the field. I.e., this non-IID split is appropriate and better serves practical algorithm development than an IID split would have. An IID split would have given an artificially high generalization estimate for what the trained model would be able to achieve "in the field". So, it is of high interest to also investigate how adaptive overfitting works in the non-IID (but realistic) cases, but in this work the non-IID "outlier" competitions are effectively put to the side. I would love to see an additional discussion (even in the appendix) on what conclusions can be drawn from this work about adaptive overfitting in non-IID cases.

[Author Response · NeurIPS 2019]

We thank the reviewers for their positive and constructive feedback. We will address all minor suggestions in the final version of the paper. We now reply to the main points raised by the individual reviewers.

**Reviewer 1**

**Point 1:** We agree with the overall conclusion of the reviewer, with the exception that the non-uniform p-values in Section 3.3 are only limited evidence for adaptive overfitting due to the strong null hypothesis. Nevertheless, some of the plots in Section 3.2 also indicate small differences between public and private accuracy scores that are indicative of mild overfitting. We will update our discussion in Section 3.5 to more clearly reflect this.

**Point 2:** We will update our figures with alpha values for the scatter plots. Figure 1 does show confidence intervals, but they are small due to the scale of the axes and the test set sizes. We will clarify this in the figure captions.

**Point 3:** Figure 4 of our submitted paper already points towards one covariate that is correlated with adaptive overfitting: in-class vs. other competitions (mainly featured competitions). This indicates that quality control performed by Kaggle for the featured competitions may avoid certain causes of public vs. private accuracy differences.

Moreover, we will include the figure on the right in the final version of our paper. The figure shows the relationship between overfitting and test set size for accuracy competitions. It also points towards 10,000 examples as a possible recommendation for minimum test set size. While a reliable recommendation for applied machine learning will require a broader investigation that goes beyond the scope of our current paper, we believe that our candidate hypothesis can still inform future work on adaptive overfitting.

**Point 4:** While we agree that ranking submissions is an important aspect of the Kaggle platform, we believe that score differences are more fundamental from the perspective of adaptive overfitting. If many models with essentially the same performance are submitted to a competition, small variations can lead to large rank changes. However, these rank changes are not problematic if the performance of each method stays the same up to statistical fluctuations. Hence we view ranking mainly as an artifact of the competition format and not inherently important for safely deploying machine learning in real applications. Having said that, we have compared our results to those of the "Meta Kaggle: Competition Shake-up" notebook on Kaggle and found similar conclusions when considering changes in ranking. We will comment on this in the final version of our paper.

**Reviewer 2**

Regarding the relationship between competition type or test set size and overfitting, please see Point 3 in our response to Reviewer 1.

We certainly plan to investigate further competition platforms and machine learning tasks (including regression). In fact, we have already contacted 20 individual ML competitions hosted outside Kaggle in order to gather data.

Regarding the ICML'19 workshop submission mentioned by Reviewer 2: please note that the corresponding workshop submission deadline was after the NeurIPS submission deadline. Due to the NeurIPS guidelines for anonymous submissions, we cannot comment on this point beyond stating that all applicable rules are being followed to the best of our knowledge.

**Reviewer 3**

We agree with the reviewer that the non-i.i.d. case (i.e., distribution shifts) is crucial for deploying machine learning and plan to investigate this point in future work. But as stated by the reviewer, the purpose of the current paper is the i.i.d. case since it allows us to clearly separate adaptive overfitting from distribution shift and show that adaptive overfitting is currently not substantively affecting machine learning practice.

Regarding lines 275–278: with "not enough data", we mean that the MetaKaggle dataset only provides the aggregate score for each submission and not the loss on each example in the test set. This makes it impossible to use techniques such as bootstrapping to compute error bounds since we cannot re-sample the test set. In contrast, the accuracy metric is simple enough so that the aggregate score is a sufficient statistic to compute tight (even exact) confidence intervals (Clopper-Pearson confidence intervals).

[Meta-Review · NeurIPS 2019]

In this paper the authors performed a large-scale empirical study of overfitting due to test set reuse in the machine learning community. The survey includes models from 112 Kaggle competitions and concluded that there is little evidence of substantial overfitting. The reviewers thought that the topic is important, that the paper was well written and easy to follow, and that the experiments are well executed. The AC and SAC had concerns that this submission might be violating the slicing-too-thin policy for NeurIPS, as described in the Call for Paper (https://neurips.cc/Conferences/2019/CallForPapers), based on similarity with another submission (5286) from an overlapping set of authors. Since these concerns appeared late in the review process (reviewers were never given a chance to comment on the submissions' similarity), ultimately the PCs reviewed the situation. They determined that the submission was sufficiently different from 5286 and could be accepted. [This meta-review was reviewed and revised by the Program Chairs]